# Laterally gated ferroelectric field effect transistor (LG-FeFET) using α-In₂Se₃ for stacked in-memory computing array

Sangyong Park[1,2,6], Dongyoung Lee[3,6], Juncheol Kang[3], Hojin Choi[3] & Jin-Hong Park [ID][3,4,5] [✉]

In-memory computing is an attractive alternative for handling data-intensive tasks as it employs parallel processing without the need for data transfer. Nevertheless, it necessitates a high-density memory array to effectively manage large data volumes. Here, we present a stacked ferroelectric memory array comprised of laterally gated ferroelectric field-effect transistors (LG-FeFETs). The interlocking effect of the α-In₂Se₃ is utilized to regulate the channel conductance. Our study examined the distinctive characteristics of the LG-FeFET, such as a notably wide memory window, effective ferroelectric switching, long retention time (over $3 \times 10^4$ seconds), and high endurance (over $10^5$ cycles). This device is also well-suited for implementing vertically stacked structures because decreasing its height can help mitigate the challenges associated with the integration process. We devised a 3D stacked structure using the LG-FeFET and verified its feasibility by performing multiply-accumulate (MAC) operations in a two-tier stacked memory configuration.

The synergies between a tremendous amount of data, artificial intelligence (AI), and data science impact almost every aspect of industries and our lives[1–3]. This trend demands a significant amount of computer resources to afford the heavy workload of the machine learning algorithm and to handle the huge data volume. As a new computing paradigm, in-memory computing enables highly efficient data-intensive computation because energy is not consumed for data migration and access[4–6]. For example, the vector-matrix-multiply (VMM) calculation, one of the most frequent operations in machine learning algorithms, can be implemented by the memory-based multiply-accumulate (MAC) operation without the data migration/access steps. The need for a high-density, low-energy, and high-precision in-memory computing array is being driven by the ongoing expansion of data volume and complexity. This, in turn, leads to an increase in the

number of input and output nodes of the memory array for in-memory computing. In-memory computing with artificial neural networks (ANN) must be developed using high-density integration technology and high-performance devices, as the size of the memory array is connected to the chip area, power consumption, and the accuracy of the MAC (multiply-accumulate) operation.

The simplest way to put more cells into the allowed area is the in-plane directional shrinking. However, as the minimum dimension limits the fabrication and performance of the device, the down-scaling technology migrates to the three-dimensional (3D) integration that exploits the vertical directional space by stacking tiers or cells instead of reducing the in-plane sizes[7]. Despite the intensive development of various vertical integration technologies, including the 3D stacked integrated circuits (3D-SICs) based on through-silicon-vias (TSVs) and

[1]Flash Technology Development Team, R&D Center, Device Solutions, Samsung Electronics Co. Ltd, Hwasung 18448, Korea. [2]Department of Semiconductor and Display Engineering, Sungkyunkwan University (SKKU), Suwon 16419, Korea. [3]Department of Electrical and Computer Engineering, Sungkyunkwan University (SKKU), Suwon 16419, Korea. [4]SKKU Advanced Institute of Nano-Technology (SAINT), Sungkyunkwan University (SKKU), Suwon, Korea. [5]Department of Semiconductor Convergence Engineering, Sungkyunkwan University (SKKU), Suwon, Korea. [6]These authors contributed equally: Sangyong Park, Dongyoung Lee. [✉]e-mail: jhpark9@skku.edu

monolithic 3D integrated circuits (M3D-ICs), several issues still hinder the further expansion to the vertical direction[8-15]. Firstly, the thermal budget for the top-layer devices is insufficient to achieve high-quality devices on back-end interconnects or even on front-end devices. Secondly, a severe increase in height as stacking requires elaboration in fabrication processes, such as anisotropic etching, flattening, alignment, and chip warpage control. In this light, thin van-der-Waals (vdW) materials are very attractive candidate materials for the implementation of 3D stacked devices[16-20]. The structure of the vdW materials is composed of atomic layers which have strong chemical bonding within the layer and weak interlayer bonding to each layer[21]. The atomic bonding structure of vdW materials facilitates the transfer of the grown film to the target substrate under low-temperature conditions[22]. As of now, the transfer technique is insufficient for industrial applications, but various transfer methods have been developed and some have succeeded in transferring a large-scale film grown by chemical vapor deposition (CVD) onto a target film[23-25]. The ultrathin thickness of vdW materials is also helpful to the 3D stacked structure because it reduces the total height and maintains surface morphology flat[26, 27]. Furthermore, the dangling-free surface affords the advantages of minimizing the roughness, traps, misfit strain, and defects in the interface[28-31]. In terms of device performance, the superior surface delivers the stable characteristics of memory devices and the immunity to mobility degradation in the ultrathin channel[32,33].

Our proposal involves a two-tier stacked device structure using various vdW materials to enable efficient MAC operation in 3D in-memory computing. Notably, we have employed a specially designed ferroelectric field-effect transistor (FeFET) as the memory device for the neural network. In contrast to conventional FeFETs, the gate electrode of our device is positioned on the side of the ferroelectric layer. The lateral gate is made functional by leveraging the unique properties of $\alpha$-In$_2$Se$_3$, namely, the interlocking feature between in-plane and out-of-plane polarizations, and the stable ferroelectric characteristics at room temperature in the ultrathin scale[34-38]. Previous studies have demonstrated the presence of ferroelectricity in a single layer (approximately 1.3 nm) of $\alpha$-In$_2$Se$_3$ and the intercoupling effect in tri-layers (approximately 3 nm)[38,39] An $\alpha$-In$_2$Se$_3$ FeFET has shown a remarkably fast ferroelectric switching time as low as 40 ns[40]. As the gate electrode is not present on the ferroelectric layer, the total height of the device structure is significantly reduced. Prior studies have employed $\alpha$-In$_2$Se$_3$ as a ferroelectric layer that is polarized by the out-of-plane (OOP) directional electric field[41-43]. Only a limited number of studies have explored the interlocking effect to alter the resistance of memristor-type devices[35,44-50]. As far as we are aware, there has been no prior exploration of the modulation of an $\alpha$-In$_2$Se$_3$ FeFET device using an in-plane (IP) directional electric field. Through experimentation, we have verified that there is an interlocking between the IP and OOP polarization in our device, and we have observed ferroelectric memory characteristics, including retention. To confirm the device profiles, we utilized transmission electron microscopy (TEM) and energy-dispersive X-ray spectroscopy (EDS). Additionally, we have verified the interlocking characteristic of $\alpha$-In$_2$Se$_3$ through piezoelectric force microscopy (PFM) and Kelvin probe force microscopy (KPFM), in terms of the phase and surface potential changes, respectively[51].

## Results

### Implementation of the laterally gated ferroelectric field-effect transistor (LG-FeFET)

A simple method to save the area without shrinking the device dimension is to stack the arrays vertically (Supplementary Note 1 and Fig. S1). To implement the vertically stacked structure, we used seven vdW layers consisting of three kinds of materials. The schematics of the stacked devices and stacking sequence of the layers are illustrated in Fig. 1a, b. The layers were stacked using the dry-transfer technique based on adhesion engineering at room temperature. $\alpha$-In$_2$Se$_3$, $h$-BN,

and MoS$_2$ were sequentially transferred twice to build the two-story memory structure, where the $\alpha$-In$_2$Se$_3$ and MoS$_2$ layers were used as ferroelectric memory layers and channels, respectively. The role of the thin $h$-BN layers between the $\alpha$-In$_2$Se$_3$ and MoS$_2$ layers is gate-dielectrics, and the thick $h$-BN layer is the interlayer-dielectric (ILD) to mitigate the interference between cells. Intriguingly, the gate electrode was formed on the side of the $\alpha$-In$_2$Se$_3$ layers to facilitate the application of the electric field in the in-plane direction. Consequently, the device has the ferroelectric-insulator-semiconductor (FIS) stack which leaves out the metal electrode from the conventional metal-ferroelectric-insulator-semiconductor (MFIS) stack, and the total height is reduced as much as the sum of the thickness of the metal electrode layers as illustrated in Fig. 1c. The optical microscope (OM) images in Fig. 1d are the top-view images of the successfully implemented device. The lower side image is the first tier and the upper side image is the second tier that was built up on the first tier. The device on the second tier was designed to have the same physical dimensions, such as channel length, width, and thickness of each layer, of the device on the first tier. The position of the second-tier's source, drain, and channel region was also aligned to the first tier. The thicknesses and the stacked profiles were confirmed by transmission electron microscope (TEM) and energy-dispersive X-ray spectroscopy (EDS) images. As shown in Fig. 1e and Supplementary Fig. S2, we confirmed that the thicknesses of the two tiers are uniform and the interfaces are clean. The ferroelectric memory characteristics of the first and second tiers were investigated with the double-sweep $I_d$–$V_g$ transfer curve as plotted in Fig. 1f. Both curves showed the counter-clockwise hysteresis, where the memory windows (MW) of the first and second tiers were 9.98 V and 8.87 V, respectively. The changes in the conductance according to positive and negative gate voltage pulses are also shown in Fig. 1g. The uniform thickness and profiles of the first- and second-tier devices resulted in a uniform range of conductance and linearity. These memory characteristics obviously indicate that the applied in-plane directional electric field successfully reverses the directions of polarization in $\alpha$-In$_2$Se$_3$ layers. The reproducibility of the ferroelectric operation was confirmed using a total of eight distinct LG-FeFET devices. Across all devices, a consistent counterclockwise hysteresis was observed in the $I_d$–$V_g$ transfer characteristic curves throughout 100 double-sweep cycles (Supplementary Fig. S3). The hysteresis loop of the drain current by the in-plane directional electric field originates from the unique atomic structure of the $\alpha$-In$_2$Se$_3$ (Se-In-Se-In-Se). The intermediate selenium atoms of the quintuple layers can move in a diagonal direction, and thereby, the IP and OOP polarizations emerge simultaneously by breaking the centrosymmetry between adjacent indium atoms as shown in Fig. 1h[34,40,41,52,53]. In other words, due to the diagonal shift of the selenium, the IP and OOP polarizations are interlocked with each other[54-57]. As a result, the in-plane directional electric field rotates the vertical directional polarization and modulates the surface potential of the channel material (MoS$_2$).

### Verification of the interlocking effect

The interlocking between the OOP and IP phases was investigated by PFM and KPFM measurement. Firstly, to confirm the rotation of the IP phase by the OOP directional electric field, a vertical electric field was applied to the ferroelectric layer ($\alpha$-In$_2$Se$_3$) through the tip of the cantilever in AFM. The center square region was scanned by positive bias (10 V) after the overall region was scanned by negative bias (−10 V). Figure 2a, b are the patterned images of the OOP and IP phases, respectively, where the OOP and IP phase images were measured simultaneously. The inversion of the phase profiles along the horizontal dashed lines in Fig. 2a, b correspond to each other at the same position as shown in Fig. 2c. The result confirmed that the OOP directional electric field, which was applied through the cantilever, changes both the OOP and the IP polarization. The local phase loop and the amplitude are in supplementary Fig. S4. Obtaining

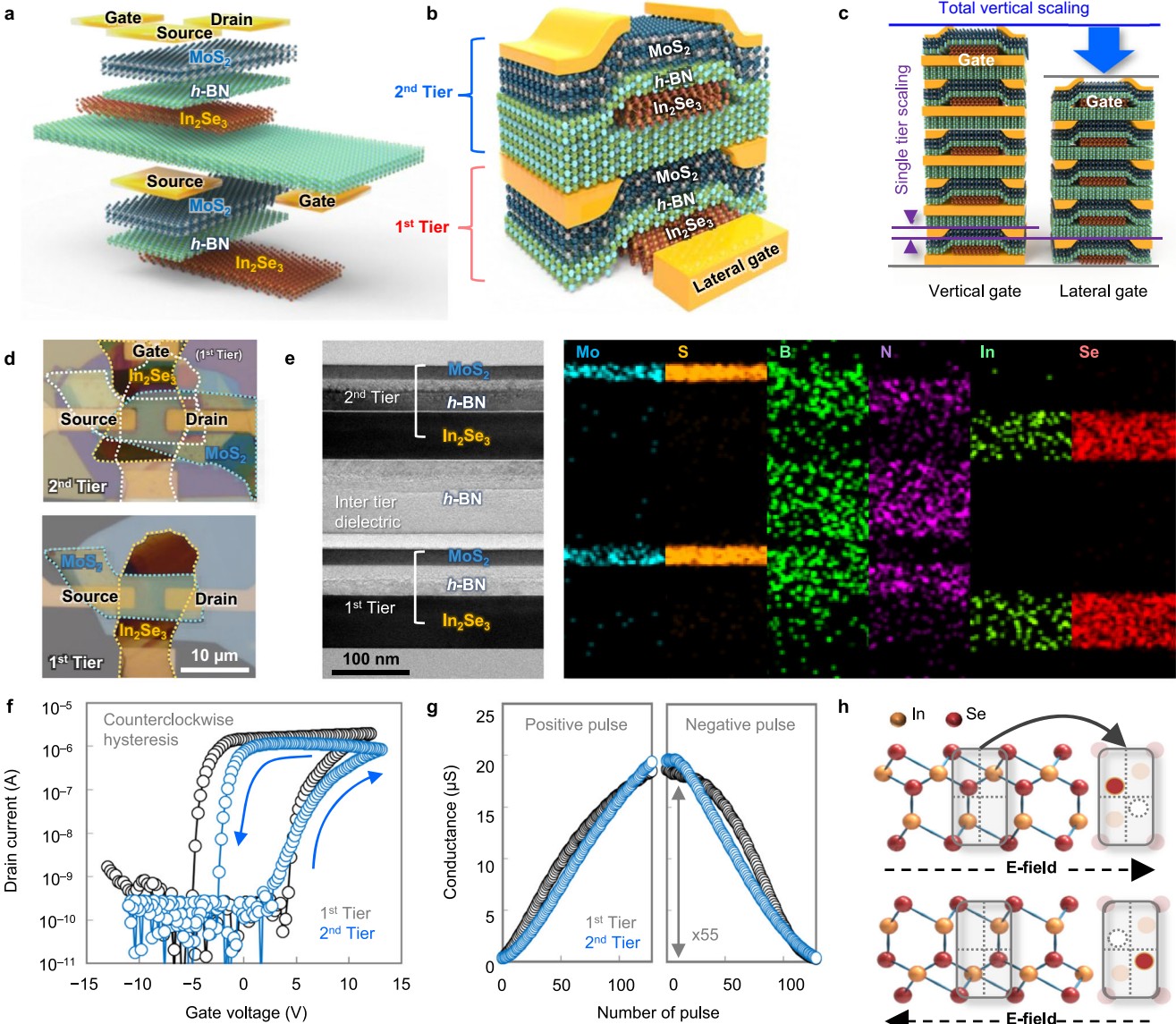

**Fig. 1 | The schematic structure and electrical characteristics of the LG-FeFET.** **a** The configuration of the layers and **b** the integrated structure of a two-tier LG-FeFET device. **c** A conceptual comparison between the LG-FeFET and a conventional vertical FeFET structure. **d** Optical microscopy images of the devices on the first and second tiers. The source and drain electrodes of the second device are aligned with those of the first device. **e** The transmission electron microscopy (TEM) image and the corresponding energy-dispersive X-ray spectroscopy in scanning transmission electron microscopy (STEM-EDS) images of the cross-section of the two-tier device. **f** Double-sweep $I_d$-$V_g$ transfer curves for both first and second tiers. The arrows in the loop indicate the sweep directions of the gate voltage. **g** The change of conductance for the incremental step positive (from −1.8 V to 0.5 V) and negative gate pulses (from −1.9 V to −4 V) with an identical read voltage (−1.8 V). **h** The atomic structures of α-In$_2$Se$_3$ (Se-In-Se-In-Se) under the external electric field.

ferroelectric properties of α-In$_2$Se$_3$ using an MFM capacitor is challenging due to its distinctive semiconducting properties (with an energy bandgap of approximately 1.3 eV). Here, we revealed a coercive voltage ($V_c$) of 1.76 V through the PFM analysis. Secondly, the modulation of the OOP polarization by the IP directional electric field was explored by the KPFM measurement. After applying the positive (5 V) and negative (−5 V) voltage to the side gate electrode while the source and drain electrodes were grounded, the surface potential was investigated by KPFM. The surface potential showed a positive potential (red color) after applying a positive voltage and a negative potential (blue color) after applying a negative voltage, as shown in Fig. 2d, e, respectively. The positive surface potential indicates that the positive charges of the upward OOP polarization are faced to the channel side due to the in-plane directional electric field, and the negative surface potential is vice versa. The electric field across the ferroelectric layer is not perfectly aligned to the in-plane direction under the channel but is

directed toward the channel because the source and drain are located on the channel layer with grounded as shown in Fig. 2f. Directions of the electric field in both positive and negative gate voltage cases are confirmed by TCAD simulation (see supplementary Fig. S5). As a result, the lateral gate electrode controls the OOP polarization, and the threshold voltage ($V_{th}$) is changed due to the interlocking effect.

## Non-volatile memory characteristics of the LG-FeFET

The memory properties of the Lateral Gated Ferroelectric Field-Effect Transistor (LG-FeFET), shown in Fig. 3a, were investigated with the $I_d$-$V_g$ transfer curves obtained by both vertical and lateral gate control on the same device. The $I_d$-$V_g$ transfer curve for the vertical gate (Fig. 3b) exhibited a counter-clockwise hysteresis loop. This is because the polarization charges on the channel side were changed by the OOP electrical field. Similarly, the transfer curve by the lateral gate control (Fig. 3c) also showed a counter-clockwise hysteresis loop, which

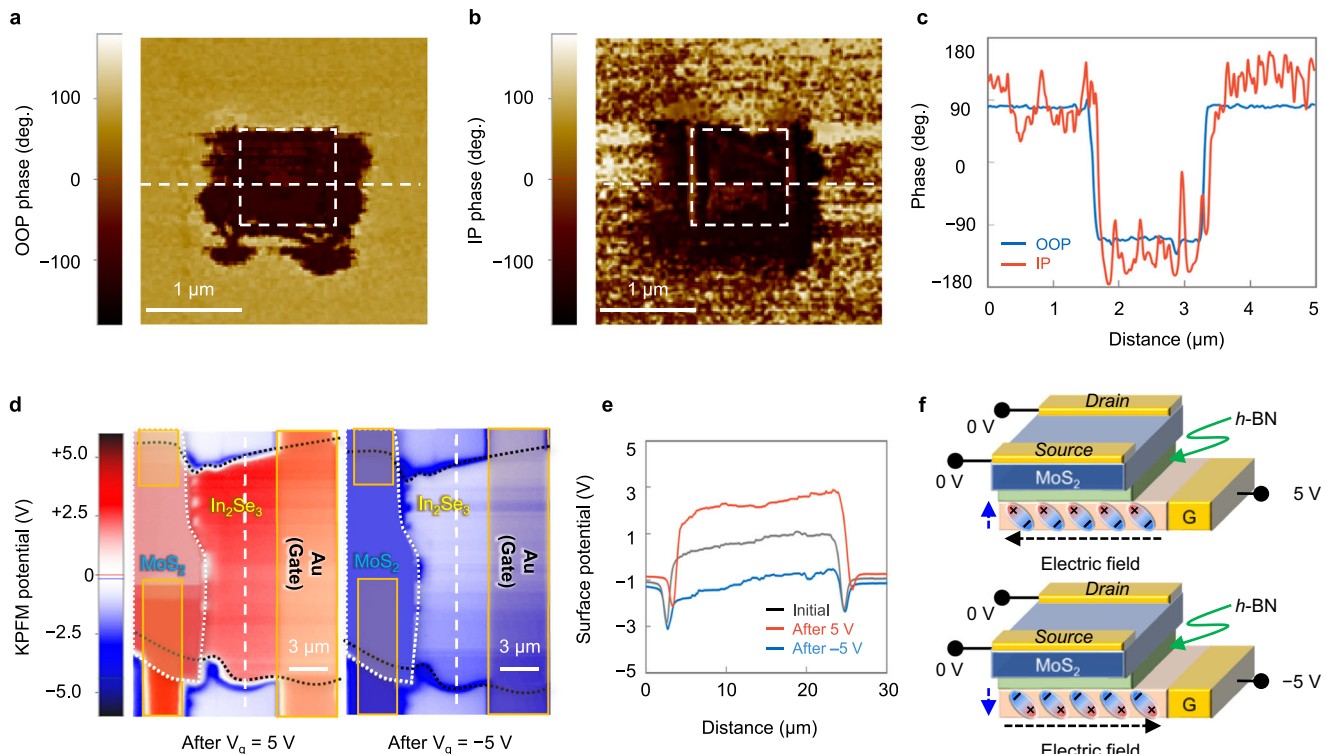

**Fig. 2 | Interlocking effect of the LG-FeFET. a** Out-of-plane (OOP) and **b** in-plane (IP) directional phase patterns observed after applying an OOP directional electric field through the tip of the cantilever in AFM. **c** Phase profiles along the white dashed lines in the phase patterns. **d** The surface potential of the LG-FeFET device after applying an IP directional electric field through the lateral gate. **e** Surface potential profiles along the white dashed lines for the initial state and after applying positive (5 V) and negative (−5 V) voltages. **f** Schematic images showing the direction of the electric dipoles in the ferroelectric layer under positive and negative voltages.

indicated that the polarization charges in the channel were controlled well by the IP electrical field. This presents the functionality of the interlocked polarization in the LG-FeFET. In this device, the interlocking effect shows minimal dependence on the orientation of the flake. Specifically, the relationship between the flake orientation and the memory window can be observed in Supplementary Fig. S6. An intriguing finding is that the memory window, represented by the width of the hysteresis loop, is wider for the lateral gate control compared to the vertical gate control. The memory window is a crucial indicator of the efficiency of ferroelectric switching. In Fig. 3d with a 24 V gate voltage range, the memory windows were 1.1 V and 7.8 V for the vertical and lateral gate control, respectively. The lateral gate control showed a significantly larger memory window, which was 6.9 times bigger than that of the vertical gate. As the voltage sweep range increased, the difference in memory windows between the two cases became even more pronounced. We observed that as the ferroelectric material thickness increases and the distance between the channel and the gate electrode decreases, the memory window tends to widen. Even at a distance of 30 μm, the memory window obtained through the lateral gate remained superior to that achieved through the vertical gate. We have included the relevant information in Supplementary Figs. S7 and S8. In Fig. 3e, the range of the gate voltage is converted to the electric field across $\alpha$-In$_2$Se$_3$ to compare polarization switching efficiency without structural effects. The electric field across the ferroelectric layer was calculated by dividing the voltage dropped in each layer by their thickness (for the vertical gate) or length (for the lateral gate). A lower electric field is needed in the IP direction to achieve the same memory window compared to the OOP direction. Furthermore, the slope of the memory window vs. electric field graph is significantly steeper in the IP direction as opposed to the OOP direction. The device's optical microscopy image and structure information can be

found in Supplementary Fig. S9. In Fig. 3f, the relationship between the electric field in the OOP and IP directions is approximately extrapolated and graphed to achieve the same memory window. In rotating the direction of the electric dipole moment, the IP directional electric field is more than 20 times as effective as the OOP directional electric field. Additionally, the ratio of the IP directional electric field to the OOP directional electric field increases as the required memory window becomes wider. By analyzing the conceptual energy diagrams presented in Fig. 3g (for the OOP directional electric field) and Fig. 3h (for the IP directional electric field), one can comprehend the results and how the electric field's magnitude impacts the activation energy barrier and total energy difference. The potential energy and polarized charge density are represented on the vertical and lateral axes, respectively. According to calculations by Ding et al., the activation energy is lower, and the total energy difference is higher under the IP directional electric field as compared to the OOP directional electric field[34]. Thus, the memory window is more sensitive to the IP directional electric field. We also note that an interlayer of SiO$_2$ is incorporated to prevent carrier injection from the vertical gate because this injection of charges compensates for the remnant polarization and decreases the memory window. We have experimentally verified this phenomenon, and the ferroelectric memory characteristics with and without interlayers are compared in Supplementary Fig. S10. On the other hand, the depolarization field in the lateral gate structure is expected to be weak due to the long distance from the gate to the channel[58]. Therefore, a large memory window can be achieved even with direct gate contact.

The OOP and IP polarization were simultaneously monitored over time to explore the intercoupling effect on the retention characteristic. The retention characteristic was evaluated using PFM, as direct measurement of the intercoupling effect in an integrated transistor is difficult, where the inner square (1×1μm$^2$) was polarized with 5 V tip

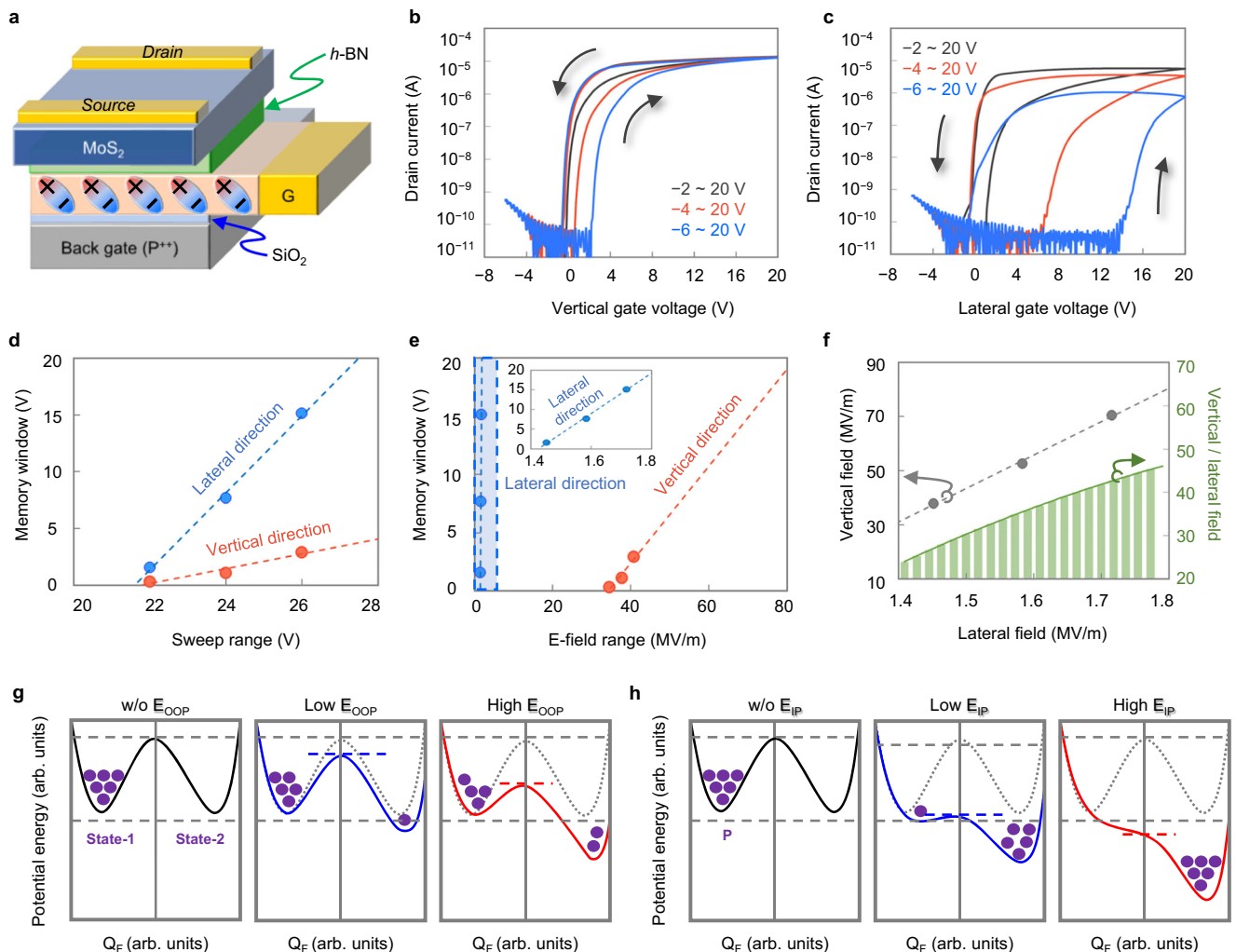

**Fig. 3 | Memory characteristics of the LG-FeFET. a** The device used to compare the memory properties by the lateral and vertical gates. **b** Double-sweep $I_d$–$V_g$ transfer curves using the vertical and **c** lateral gates. **d** Comparison of memory windows between the vertical and lateral gates as a function of the sweep range and **e** the electric field across the ferroelectric layer. **f** Correlation chart showing the relationship between the lateral and vertical directional electric fields within the

same memory window. The bar chart shows the ratio of the vertical and lateral electric fields. **g** and **h** are conceptual energy diagrams that explain the difference of memory windows and the magnitude of the electric field. The energy diagrams are based on the Landau-Devonshire theory and first-principle density functional theory calculation, with respect to **g** the OOP and **h** the IP directional electric fields[34, 66].

voltage after applying −5 V to the outer area. Figure 4a, b show the images of the OOP and IP phases at different time points, respectively. The interlocked polarizations of the α-In$_2$Se$_3$ resulted in polarized phase patterns in both OOP and IP directions, which remained distinguishable for a duration of over $10^4$ s. Figure 4c provides a quantitative representation of the changes in the PFM images as a retention characteristic. Up to $3 \times 10^4$ s, the OOP and IP polarizations remained in their initial states. However, after this time, the polarization decreased by 10 percent over the next $10^4$ s, while the patterns' boundaries became blurred as depicted in Fig. 4a, b. The degradation of the ferroelectric retention property occurs in both the inner square (polarized to negative phase) and the outer region (polarized to positive phase). To gain a comprehensive understanding of the retention behavior of α-In$_2$Se$_3$, we examined the transient distributions of pixels for the phases in Fig. 4d. As time progressed, the polarized phases gradually shifted towards the opposite polarity from their initial polarity, while the distributions spread. The statistical mean values and the standard deviations of the distributions are illustrated in Fig. 4e, f, respectively. It is worth mentioning that the speed of the phase change is slower in the IP polarization compared to the OOP polarization. The phase profiles across the patterns were compared in Fig. 4g to analyze

the phase transitions at the boundary. It can be observed that the phase change in the OOP direction is uniform across all regions without any boundary shift. In contrast, in the IP direction, the phase change mostly occurs at the boundary and propagates towards the center. The difference in retention behavior between the OOP and IP polarizations can be attributed to the neighboring dipole charges at the boundary. The reason for this is that the electric dipoles' configuration in the IP and OOP phase patterns has different discontinuous arrangements at the boundary, as illustrated in Fig. 4h. The dipole charges of the IP directional component face charges of the same polarity on the opposite side of the boundary. Conversely, the dipole charges of the OOP directional component align with charges of opposite polarity on the opposite side of the boundary. This affects the depolarization field differently: the same polarity charges aligned in the IP direction strengthen the depolarization field at the boundary, while the opposite polarity charges aligned in the OOP direction weaken the depolarization field[59]. As a result, the depolarization field in the IP direction causes electric dipoles to rotate, resulting in the phase boundary shifting inward. On the other hand, the depolarization field in the OOP direction maintains the boundary. The OOP and IP directional polarization exhibit an imperfect correlation during retention

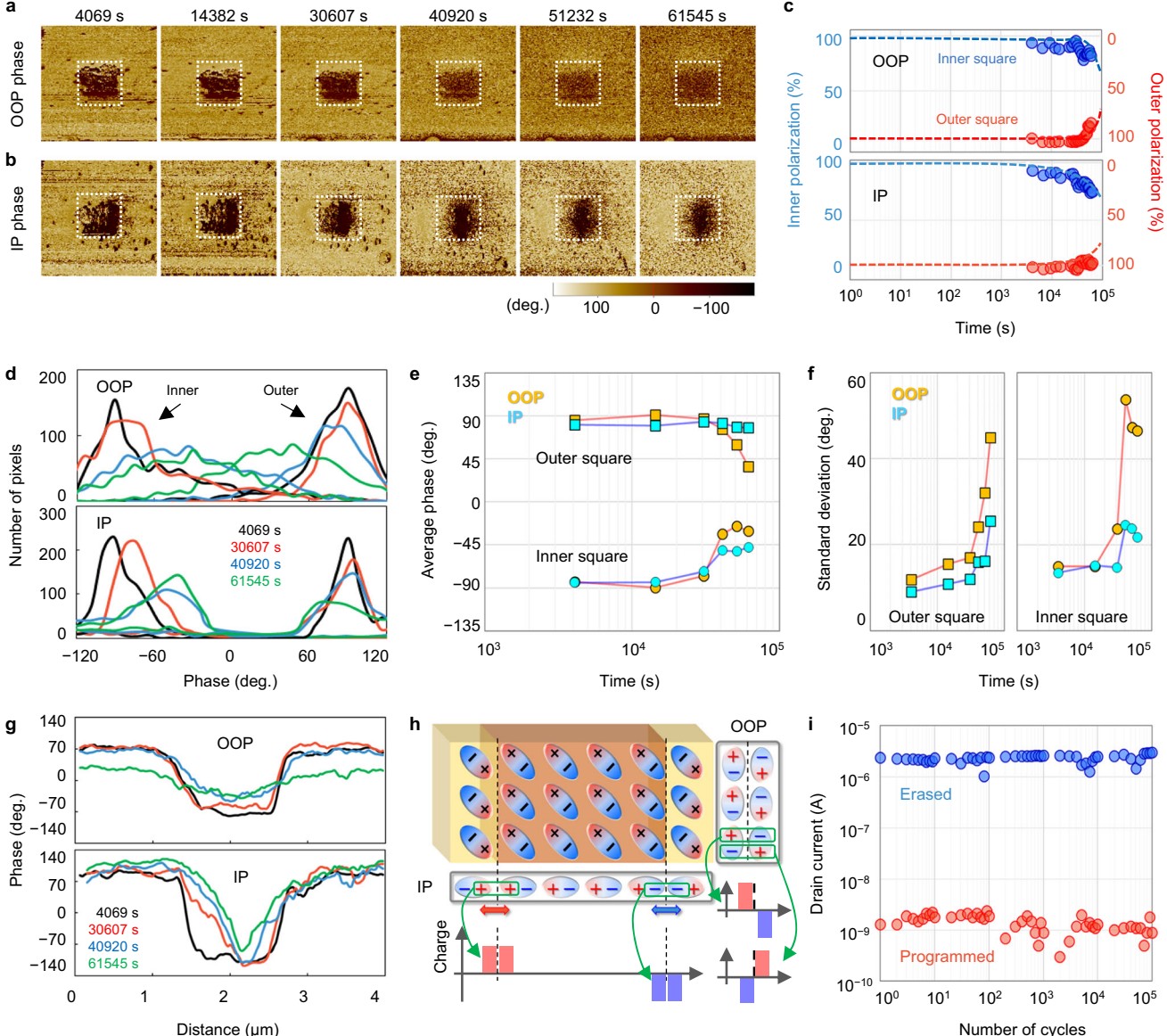

**Fig. 4 | Reliability characteristics of the LG-FeFET. a** and **b** are the PFM images of the OOP and IP phase patterns, respectively, during the retention time. **c** The change of the OOP and IP phases in the inner square and outer region. The size of both areas is $1 \times 1\,\mu m$. The portions of each polarization are calculated by counting the number of pixels for each polarity. **d** The number of pixels for each phase during the retention time. **e** The average values of each distribution and **f** standard deviation during the retention time. **g** Transient profiles of the phases across the inner square and outer region. **h** The dipole configuration and the net polarization charge at the boundary. **i** The drain current for the number of program-erase cycles.

time, despite their intercoupling effect. To understand this phenomenon, the impact of neighboring dipoles, activation barriers, and variations in total energy need to be investigated. Supplementary Table 1 lists the retention characteristics of $\alpha$-In$_2$Se$_3$ ferroelectric memory, as reported by several studies[35, 44,47,55,60]. As far as we know, this study is the first to comprehensively investigate both the IP and OOP directional retention characteristics of $\alpha$-In$_2$Se$_3$ ferroelectric memory, while also considering the intercoupling effect between polarizations. The LG-FeFET, as shown in Supplementary Fig. S11, is expected to exhibit a shorter retention time compared to that predicted on individual flakes through PFM analysis. Several factors, including measuring conditions, neighboring layers, and defects, can account for this difference. The LG-FeFET device's endurance was assessed using the lateral gate configuration. Ferroelectric switching was successfully performed over $10^5$ cycles, with no gradual degradation caused by carrier trapping or ferroelectric material fatigue. During the endurance test, the program and erase pulses were applied with amplitudes of 4 V and −7 V,

respectively. After each pulse, the programmed/erased states were verified under a read voltage of −0.7 V. The pulse rate for the endurance test was set at 0.1 kHz. Figure 4i confirms the consistent current of the programmed (low conductance) and erased (high conductance) states. We briefly compared the features of LG-FeFET and HZO-based FeFET in Supplementary Table 2. The LG-FeFET exhibits a larger memory window (approximately 10 V) compared to HZO-based FeFETs (<5 V)[61]. The endurance of LG-FeFET is comparable with HZO FeFETs, but the retention is shorter than that of HZO FeFETs. Due to the unique semiconducting properties of $\alpha$-In$_2$Se$_3$, the coercive field ($E_c$) and the remnant polarization ($P_r$) cannot be directly compared.

**Stacked LF-FeFET array for in-memory computing**

The LG-FeFET has distinct benefits in reducing the vertical height and regulating weight values since there is no metal gate and a considerable memory window. To increase storage density and lower energy consumption, we suggest a 3D stacked design that can utilize the

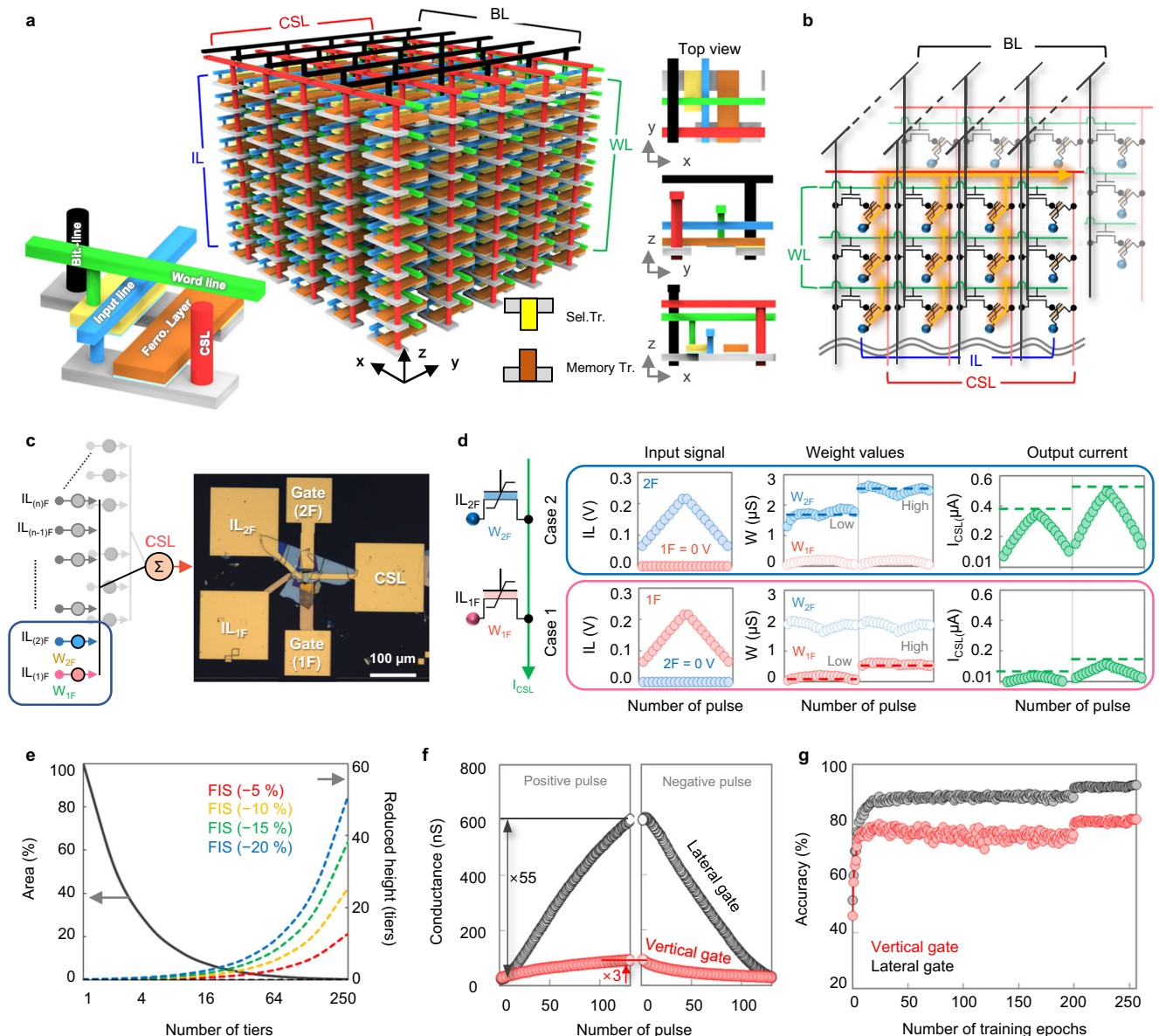

**Fig. 5 | A vertically stacked 3D in-memory computing array structure and its applications. a** A vertically stacked 3D in-memory computing memory structure composed of LG-FeFETs. The top view and side views are displayed in the right panel. **b** The equivalent circuit diagram corresponding to the structure shown in **a**. **c** The OM image of the two-tier LG-FeFET device for the MAC operation. **d** The results of the MAC operation on the two-tier LG-FeFET device for the various input signals and weight values. **e** The reduction of the area and height as a function of the number of tiers at the same density. **f** The comparison of the dynamic range between the vertical and lateral gates. **g** The results of the image recognition simulation, which includes the dynamic ranges of the vertical and lateral gates, with a convolution neural network (CNN) simulation for the CIFAR-10 dataset.

exceptional characteristics of the LG-FeFET. In Fig. 5a, we display the comprehensive schematic and cross-sectional images of the proposed 3D stacked structure. The unit cell of this 3D structure is made up of a selection transistor and a memory transistor. The selection transistor's primary function is to dictate the behavior of the memory transistor through a combination of signals from the bit-line (BL) and word-line (WL). These signal lines connect to the drain and gate of the selection transistor, respectively. On the other hand, the memory transistor is responsible for carrying out the multiplication during the MAC operation. An input signal in the form of voltage is accepted through the input line (IL) and is then transmitted through the channel. As per Ohm's law, this voltage is converted into a current and sent out to the common source line (CSL). The CSL and BL vertically connect the unit cells along the z-axis, while groups of cells are connected in parallel along the y-axis. A 'MAC plane' is defined as a group of cells that are

connected by the same CSL on the y-z-plane, and this serves as the fundamental unit for performing MAC operations. The total number of MAC planes required is dependent on the number of output nodes. The corresponding equivalent circuit is shown in Fig. 5b. More detailed operation scheme of MAC planes is described in Supplementary Table 3. Input data is received by the MAC plane via the IL and a solitary output current is generated through the CSL (Fig. 5c). In order to demonstrate the feasibility of the 3D stacked memory structure utilizing LG-FeFETs, we carried out an experimental validation of MAC operation in two stacked LG-FeFET devices (Fig. 5c). As shown in Fig. 5d, two LG-FeFET devices are linked to the CSL line, from which the output currents come out. Different input voltages were applied to the 1F and 2F ILs of LG-FeFET devices, each with distinct weight values, resulting in different CSL currents. The currents correspond to the product of the input signal amplitudes and the respective weight

values. Subsequently, these currents are accumulated at the CSL, following Kirchhoff's law, as the sources of the two LG-FeFET devices are interconnected at a shared node. Refer to the two cases in Fig. 5d. The efficiency of the vertically stacked structure is contingent on the number of tiers. As the number of tiers is raised while keeping the density the same, less chip area is necessary, although the fabrication processes for vertical interconnection become more intricate at the same time[62, 63]. Fortunately, the 3D structure employing the LG-FeFET can alleviate the level of difficulty by relocating the gate electrode regardless of whether they are stacked in a sequential or alternative manner, which reduces the overall height. The reduction in area and total height is shown as a function of the number of stacks in Fig. 5e. Scaling the LG-FeFET vertically results in a reduction of the total height, which consequently leads to a decrease in both vertical directional resistance and energy consumption during a read operation. More details about the energy consumption and vertical resistance based on the number of tiers can be found in Supplementary Fig. S12. In comparison to a conventional vertical gate memory device, the LG-FeFET has exhibited a much larger memory window, which implies that the device's channel conductance varies significantly over a wide range at a read gate voltage. Figure 5f illustrates the contrast in the conductance's dynamic ranges between the vertical gate and lateral gate on the same device. To obtain the linear change of the conductance, the incremental step pulses with identical read voltage were applied to both vertical and lateral gates. The incremental step pulse program/erase (ISPP/ISPE) conditions were separately optimized for the vertical and lateral gates. For the ISPP condition, we set the start voltage at 2.3 V and the stop voltage at 4.0 V, with an increment of 13 mV. The ISPE condition, on the other hand, had a start voltage of −3.0 V and a stop voltage of −4.0 V, with an increment of 8 mV. Both pulse rates were maintained at 1 kHz. The states were verified at a gate voltage of 0.7 V after each program/erase pulse. The lateral gate of LG-FeFET exhibits a significantly larger dynamic range of conductance, measuring 55, which is 18 times larger than the vertical gate's range of 3. The LG-FeFET is particularly advantageous in achieving more precise MAC (multiply-accumulate) operations due to its ability to handling a broad range of weight values. To evaluate the improved performance of LG-FeFET in system level, we conducted an image recognition simulation using a convolutional neural network (CNN) and the CIFAR-10 dataset[64,65]. As displayed in Fig. 5g, the lateral gate achieved an accuracy of 92.6%, whereas the vertical gate remained at 80.4%.

## Discussion

We have successfully developed a two-tier stacked ferroelectric memory that takes advantage of the distinctive material properties of α-In$_2$Se$_3$. The memory device is constructed using a laterally gated FeFET structure, wherein the gate is positioned on a section of the ferroelectric layer to apply an IP directional electric field. The primary operational mechanism of the LG-FeFET is based on the interlocking effect between the IP and OOP polarizations, which has been verified through PFM and KPFM measurements. Based on our experimental findings, we have confirmed that the IP directional electric field was 20 times more efficient in polarizing switching than the OOP directional electric field. As a result, the LG-FeFET demonstrated a broader memory window than the vertically gated FeFET. Additionally, we examined the retention characteristics of both IP and OOP polarizations simultaneously. Both polarizations maintained their polarized states for a duration exceeding $3 \times 10^4$ s. Nevertheless, we have noted that the two polarizations exhibited distinct changing behaviors during the retention period. We have observed that the IP polarization degraded from the boundary, while the OOP polarization degraded across the entire pattern. Using stacked LG-FeFETs, we have created an in-memory computing array and have successfully performed a MAC operation with a two-tier stacked memory. The wide dynamic range of LG-FeFETs

provided enhanced accuracy for computing based on an ANN compared to the vertical gate structure.

## Methods

### Fabrication of the LG-FeFET

The substrate is a heavily p-doped silicon wafer with a thermally grown 90 nm thick SiO$_2$ layer. The α-In$_2$Se$_3$ (2H), h-BN, and MoS$_2$ flakes were mechanically exfoliated in turn from their bulk crystals using an adhesive tape (224SPV, Nitto). The α-In$_2$Se$_3$, h-BN, and MoS$_2$ flakes were transferred onto a polydimethylsiloxane (PDMS) layer, and sequentially, transferred onto the SiO$_2$ surface using a dry-transfer machine. The α-In$_2$Se$_3$ flake was partially covered with the h-BN flake and the channel material (MoS$_2$) was placed on the overlapped region of α-In$_2$Se$_3$ and h-BN. Electron beam lithography (EBL) was used to define the gate, drain, and source regions. EBL photoresists, poly methyl methacrylate (PMMA) A4 and A6, were spin-coated at 3000 rpm for 60 s and baked at 180 °C for 120 s. The gate was defined on the opened region of the α-In$_2$Se$_3$ flake including the edge. After forming the PMMA pattern, Ti/Au (10 nm/80 nm) were deposited using an e-beam evaporator. The outside of the defined metal electrode regions was removed by the lift-off process. Before stacking another LG-FET device, a thick interlayer dielectric (ILD) was transferred onto the first device to separate the tiers electrically. The aforementioned LG-FET process was repeated to fabricate the second-tier device. To achieve consistency between the first and second tiers, we employed a process of selecting exfoliated flakes based on their color classification. Furthermore, we confirmed the thickness of these flakes using AFM measurements. This approach ensured uniformity in our samples.

### Characterization of the materials and devices

The surface morphologies of the devices were inspected by AFM measurement using a non-contact cantilever with a high resonant frequency (PPP-NCHR, nanosensor) probe. The OOP and IP phases of the α-In$_2$Se$_3$ were inspected by PFM measurement using a conductive tip coated with Cr-PtIr5 (PPP-CONTSCPt, nanosensor). The analysis of the surface potential was conducted via KPFM measurement using a conductive tip coated with Cr-Au (NSC14/Cr-Au, Mikromasch). The AFM, KPFM, and PFM analyses were performed using an NX10 system (Park Systems Corp.). The current-voltage characteristics were investigated using Keysight B2912a, B2902a, and B1500a semiconductor parameter analyzers. All the measurements were conducted at room temperature in the air.

## Data availability

All data that support the findings of this study are included in the article and the Supplementary Information file. These data are available from the corresponding author upon request.

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

## Acknowledgements

This research was supported by the National Research Foundation of Korea (NRF) (2022M3F3A2A01072215, 2022M3H4A1A04096496).

## Author contributions

S.P. and D.L. designed the experiments and analyzed the data. S.P. designed the 3D structure and operation scheme. D.L. and J.K. performed the AFM, KPFM, and PFM measurements. D.L. and H.C. fabricated memory devices and performed electrical measurements. J.-H.P. supervised the research. All authors have discussed the results and commented on the manuscript.

## Competing interests

The authors declare no competing interests.
