## [Peer Review File · Nature Communications]

REVIEWER COMMENTS

Reviewer #1 (Remarks to the Author):

In this paper, the authors demonstrated the distinctive characteristics of the LG-FeFET, showing this device is well-suited for implementing vertically stacked structures. In addition, the authors devised a 3D stacked structure using the LG-FeFET and verified its feasibility by performing MAC operations in a two-tier stacked memory configuration. This work is interesting; however, I have major and minor comments in its current form.

Major Comments:

1. Could the authors please provide more detailed information on how they controlled the uniformity of the flakes' thickness? As seen in Figure 1e, the flakes on the first and second tiers are notably uniform.
2. The authors utilized the interlocking property of α -In₂Se₃ to design the LG-FeFET, but it's possible that the orientation of the flake was randomly transferred, resulting in a random in-plane directional orientation of the gate. Could the authors provide experimental or theoretical information regarding the dependence on the orientation of α -In₂Se₃?
3. The authors demonstrated in Figure 3e that the in-plane directional electric field is more effective than the out-of-plane directional electric field, likely due to the longer distance between the gate electrode and the channel. If the length of the ferroelectric layer between the channel and the gate electrode is varied, it may impact the memory window. Can the authors provide information on how the length between the channel and the gate electrode affects the memory window?
4. In Figure 4d, the authors showed that the lateral gate had a larger dynamic range than the vertical gate when using incremental step pulses. It would be beneficial if the authors provided a detailed description of the measuring method in the Methods section or another appropriate section.
5. In Figure 4g, there appears to be imperfect correlation between the IP and OOP polarizations, suggesting that the interlocking effect of α -In₂Se₃ does not result in a perfect one-to-one matching between the IP and OOP polarizations during the retention time. This observation should be explicitly mentioned in the main article.

Minor Comments”

1. On page 7, there is a typo where "remanent" should be replaced with "remnant".
2. It is possible that "Fig. 1a" on page 8 is intended to refer to "Fig. 4a".
3. In Figure 5(d), it appears that W2F, which represents the weight of the second layer, should be corrected to W1F.

Reviewer #2 (Remarks to the Author):

This paper investigates the laterally gated ferroelectric field effect transistor based on α -In₂Se₃. The authors exploit the interlocking effect between the in-plane and out-of-plane polarization in α -In₂Se₃ to implement the FeFET. It leverages the unique properties of the α -In₂Se₃ and demonstrates interesting device characteristics. Several material level characterizations are performed understand the interlocking of in-plane and out-of-plane polarization and the retention of polarization. Using a small demonstration, the authors demonstrate the possibility of applying the lateral FeFET for the compute in memory applications. Though the idea is novel, the interpretation of the data is not very convincing and require clear explanation.

1. It is not very convincing regarding the need of stack height reduction and the use of vdW materials. The reviewer agrees that is a critical issue for nowadays 3D memory, for example vertical NAND. However, what the authors propose in this work is a sequential 3D process, as shown in Fig.5. With layer by layer processing, it is less obvious why the stack height with a vertical metal electrode could be a serious issue. Besides, this work is far from the vdW vision that has a stack height of 100s of nm. The authors are challenged to provide a better argument for lateral FeFET given that the lateral FeFET will likely have a larger footprint.

2. The comparison between the lateral and vertical gate is not very convincing. The reason why a much less window is observed in vertical gate than the lateral gate is unclear. The authors have spent much efforts explaining that the lateral electric field is much efficient in switching the polarization. But how that is translated to a larger window? What (the design parameters of the device) controls the lateral FeFET memory window? The authors mentioned in Fig.3e, the gate voltage is converted to electric field. How that conversion is done? For the vertical FeFET, the insertion of SiO₂ layer does not necessary reduce the memory window, which could actually

increase the window as the required switching voltage increases. Fig.3g and h are very handwaving, without much physical ground.

3. How reproducible are the reported characteristics of the device? The authors are required to show the multi cycle sweep on multiple devices.

4. For the retention measurement, it is unclear why there is no direct transistor state retention measurement, but rather the PFM. The authors need to show the transistor measurement results.

5. A big issue with this work is the lack of experimental details regarding the electrical measurements. Are all the measurements done in DC? What is the switching time of the device? How about the gate leakage current? What are the write pulses for the synaptic demonstration and endurance measurement?

6. How to make sense of the Fig.5d is unclear. Whether a linear result is obtained for MAC is unclear. The middle panel for Fig.5d should be W1F.

7. The authors need to discuss about the scalability of the lateral FeFET.

Reviewer #3 (Remarks to the Author):

This paper investigates the laterally gated FeFET using alpha-In₂Se₃ as the ferroelectric, the interlocking of IP and OOP, the memory characteristics of the device and finally its application for in-memory computing. The work is original, interesting, scientifically sound and I recommend it for publication. Here are my comments:

1) Could you briefly elaborate on the size effect of the interlocking effect, i.e. what happens when the vertical and lateral size of the ferroelectric (FE) layer change? E.g. one would expect that the effect of the lateral gate is stronger for thinner and laterally shorter FE layers.

2) If this is the case, does this pose a limitation for the future device size and architectures? Because, it is well known that the thicker the FE layer, the larger the memory window (e.g. check [R1] on this). However, if there is a maximum thickness above which the lateral gate cannot switch the polarization and/or the interlocking is not efficient, this might be a problem.

3) On the other hand, the FE layer is relatively thick (60-70nm) as compared to the sizes of highly scaled memory devices in the semiconductor industry. Can this be further scaled down? What is the prospect?

4) What are the main ferroelectric properties of the FE layer (e.g. coercive field, polarization etc)? Could you please provide this in the main text?

5) Fig. 4(i) shows the cycling endurance. What amplitude/pulse duration was applied? Please provide in the main text. In general, how does the device behave under pulsed operation? What is the switching time?

6) The memory window, larger than 9V, as well as the retention and the endurance of this device are remarkable. However, it would be quite useful for the ferroelectrics/memory community to have a comparison with state-of-the art FeFETs, which are currently mainly made of Hafnium-(Zirconium)-Oxide. I suggest to provide a brief comparison in the main text, e.g., in terms of the main device and material properties summarized in [R1].

[R1] Mulaosmanovic, H., et al. "Ferroelectric field-effect transistors based on HfO₂: a review." *Nanotechnology* 32.50 (2021): 502002.

Point-by-Point Responses to the Reviewer Comments

Reviewer #1 (Remarks to the Author):

In this paper, the authors demonstrated the distinctive characteristics of the LG-FeFET, showing this device is well-suited for implementing vertically stacked structures. In addition, the authors devised a 3D stacked structure using the LG-FeFET and verified its feasibility by performing MAC operations in a two-tier stacked memory configuration. This work is interesting; however, I have major and minor comments in its current form.

Author response: Thank you for reviewing our paper. We appreciate your insightful comments on our research. We have revised the manuscript according to your suggestions and believe that these revisions have helped improve the paper.

Please find below our responses (in blue) to each of your specific comments (in black). Revisions to the original article are indicated in red.

Major Comments:

Comment 1: Could the authors please provide more detailed information on how they controlled the uniformity of the flakes' thickness? As seen in Figure 1e, the flakes on the first and second tiers are notably uniform.

Author response 1: Controlling the thickness of flakes during the exfoliation process proves challenging in practice. In order to achieve consistency between the first and second tiers, we employed a meticulous selection process for the flakes. Subsequently, we verified the thickness of all flakes by conducting AFM measurements after their classification according to color. In the Method section, we mentioned details regarding the strict control of optical microscope brightness and the maintenance of consistent environmental conditions in the laboratory. These measures were implemented to ensure accurate classification of the samples.

In the main article:

“...The aforementioned LG-FET process was repeated to fabricate the second-tier device. To achieve consistency between the first and second tiers, we employed a process of selecting exfoliated flakes based on their color classification. Furthermore, we confirmed the thickness of these flakes using AFM measurements. This approach ensured uniformity in our samples.”

Comment 2: The authors utilized the interlocking property of α -In₂Se₃ to design the LG-FeFET, but it's possible that the orientation of the flake was randomly transferred, resulting in a random in-plane directional orientation of the gate. Could the authors provide experimental or theoretical information regarding the dependence on the orientation of α -In₂Se₃?

Author response 2: Thank you for your valuable comments. To investigate the impact of the in-plane directional orientation of α -In₂Se₃, we utilized a multiple lateral gated FeFET device, as depicted below. While the orientation of each gate with respect to the channel may be random, the gates adequately encompass a broad range of directions surrounding the channel. The I_d-V_g transfer curves displayed below exhibit the presence of an interlocking effect across all gates, with the device characteristics showing minimal dependence on the gate orientation. We have included the relevant information in Supplementary Figure S6 and the main article.

“Supplementary Figure S6 a) Optical microscopy image of the device which has eight gates around the channel region. b) I_d - V_g transfer curves for the gates, which are swept from -5 V to 10 V. All curves show counterclockwise hysteresis and hardly depends on the direction of the gates. c) Memory windows for the various gate directions. The memory windows are extracted at the 1 nA drain current.”

In the main article:

“...the interlocked polarization in the LG-FeFET. In this device, the interlocking effect shows minimal dependence on the orientation of the flake. Specifically, the relationship between the flake orientation and the memory window can be observed in Supplementary Fig. S6.”

Comment 3: The authors demonstrated in Figure 3e that the in-plane directional electric field is more effective than the out-of-plane directional electric field, likely due to the longer distance between the gate electrode and the channel. If the length of the ferroelectric layer between the channel and the gate electrode is varied, it may impact the memory window. Can the authors provide information on how the length between the channel and the gate electrode affects the memory window?

Author response 3: We appreciate your insightful feedback. We conducted an investigation on the memory window, exploring various distances between the channel area and the gate electrode. Our results indicate that with increasing distance, the memory window diminishes due to a decrease in the applied e-field across the ferroelectric layer. It is worth noting that despite the distance ($30 \mu\text{m}$) being considerably greater than the thickness of the ferroelectric layer ($60\sim 70 \text{ nm}$), the memory window achieved through the lateral gate remains larger compared to that obtained through the vertical gate. The relevant details have been included in Supplementary Fig. S8 and the main article.

“Supplementary Figure S8 a) Optical microscopy image of the LG-FeFET device which has three gates. b) I_d - V_g characteristic curves for the vertical and lateral gates and c) the extracted memory windows. The lateral gates are positioned at distances of 10 μm , 20 μm , and 30 μm away from the channel area, respectively. All curves show counterclockwise hysteresis.”

In the main article:

“...the difference in memory windows between the two cases became even more pronounced. We also observed that as the thickness of the ferroelectric material and the distance between the channel and the gate electrode increased, the memory window decreased. Interestingly, even at a distance of 30 μm , the memory window obtained through the lateral gate remained superior to that achieved through the vertical gate. Supplementary Fig. S8 provides relevant data on the impact of the ferroelectric layer's thickness and length on memory windows.”

Comment 4: In Figure 4d, the authors showed that the lateral gate had a larger dynamic range than the vertical gate when using incremental step pulses. It would be beneficial if the authors provided a detailed description of the measuring method in the Methods section or another appropriate section.

Author response 4: We conducted an optimization of the incremental step pulse program/erase (ISPP/ISPE) for both the vertical and lateral gates. For ISPP, the condition was set to start at 2.3 V and stop at 4.0 V, with an increment of 13 mV. Similarly, for ISPE, the condition was set to start at -3 V and stop at -4 V, with an increment of 8 mV. The pulse rates for both operations were maintained at 1 kHz. After each program/erase pulse, the states were verified at a gate voltage of 0.7 V. We have included these details in the main article.

In the main article:

“...both vertical and lateral gates. The incremental step pulse program/erase (ISPP/ISPE) conditions were individually optimized for both the vertical and lateral gates. For ISPP, the condition was set to start at 2.3 V and stop at 4.0 V, with an increment of 13 mV. Similarly, for ISPE, the condition was set to start at -3 V and stop at -4 V, with an increment of 8 mV. The pulse rates for both operations were maintained at 1 kHz. After each program/erase pulse, the states were verified at a gate voltage of 0.7 V. We have included these details in the main article.”

Comment 5: In Figure 4g, there appears to be an imperfect correlation between the IP and OOP polarizations, suggesting that the interlocking effect of α - In_2Se_3 does not result in a

perfect one-to-one matching between the IP and OOP polarizations during the retention time. This observation should be explicitly mentioned in the main article.

Author response 5: We appreciate the reviewer's valuable feedback. One of the interesting observations we made is the lack of a perfect correlation between the OOP and IP directional polarization during retention time, despite their clear intercoupling effect. To understand this phenomenon, we took into account the impact of neighboring dipoles, activation barriers, and variations in total energy. However, further detailed investigations are required to gain a comprehensive understanding. Since this exceeds the scope of the current paper, we have included the relevant information in the main article and left it as a topic for future research.

In the main article:

“...in the OOP direction maintains the boundary. The OOP and IP directional polarization exhibit an imperfect correlation during retention time, despite their intercoupling effect. To understand this phenomenon, the impact of neighboring dipoles, activation barriers, and variations in total energy need to be investigated.”

Minor Comments

1. On page 7, there is a typo where "remanent" should be replaced with "remnant".
2. It is possible that "Fig. 1a" on page 8 is intended to refer to "Fig. 4a".
3. In Figure 5(d), it appears that W2F, which represents the weight of the second layer, should be corrected to W1F.

Author response: Thank you for your detailed comments. We modified all the minor comments in the revised manuscript.

Reviewer #2 (Remarks to the Author):

This paper investigates the laterally gated ferroelectric field effect transistor based on α -In₂Se₃. The authors exploit the interlocking effect between the in-plane and out-of-plane polarization in α -In₂Se₃ to implement the FeFET. It leverages the unique properties of the α -In₂Se₃ and demonstrates interesting device characteristics. Several material level characterizations are performed understand the interlocking of in-plane and out-of-plane polarization and the retention of polarization. Using a small demonstration, the authors demonstrate the possibility of applying the lateral FeFET for the compute in memory applications. Though the idea is novel, the interpretation of the data is not very convincing and require clear explanation.

Author response: Thank you for reviewing our paper. We appreciate your insightful comments and revised the manuscript accordingly.

Please find below our responses (in blue) to each of your specific comments (in black). Revisions to the original article are indicated in red.

Comment 1: It is not very convincing regarding the need of stack height reduction and the use of vdW materials. The reviewer agrees that is a critical issue for nowadays 3D memory, for example vertical NAND. However, what the authors propose in this work is a sequential 3D process, as shown in Fig.5. With layer-by-layer processing, it is less obvious why the stack height with a vertical metal electrode could be a serious issue. Besides, this work is far from the vdW vision that has a stack height of 100s of nm. The authors are challenged to provide a better argument for lateral FeFET given that the lateral FeFET will likely have a larger footprint.

Author response 1: We appreciate your valuable feedback. The gate metal electrodes play a crucial role in increasing the height of the memory array, regardless of whether they are stacked in a sequential or alternative manner. Consequently, this can pose challenges in forming a common source electrode (vertical interconnect line). The reason behind this difficulty lies in the simultaneous establishment of the source electrodes, which needs to be done through the edge of the MoS₂ channel layers to minimize costs and ensure accurate alignment.

By 2030, industries anticipate the advancement of vertical stacking technology to surpass a thousand layers. This prediction suggests that vertical scaling down is unavoidable to address resistance reduction and tackle processing challenges linked to vertical interconnect lines. In this regard, van der Waals (vdW) materials emerge as the most favorable contenders for decreasing the vertical dimension owing to their atomic-scale thickness. We have incorporated this pertinent information into the main article.

In the main article:

“Fortunately, the 3D structure employing the LG-FeFET can alleviate the level of difficulty by relocating the gate electrode **regardless of whether they are stacked in a sequential or alternative manner**, which reduces the overall height.”

Comment 2: The comparison between the lateral and vertical gate is not very convincing. The reason why a much less window is observed in vertical gate than the lateral gate is unclear. The authors have spent much efforts explaining that the lateral electric field is much efficient in switching the polarization. But how that is translated to a larger window? What (the design parameters of the device) controls the lateral FeFET memory window? The authors mentioned in Fig.3e, the gate voltage is converted to electric field. How that conversion is done? For the vertical FeFET, the insertion of SiO₂ layer does not necessary reduce the memory window, which could actually increase the window as the required switching voltage increases. Fig.3g and h are very handwaving, without much physical ground.

Author response 2: We appreciate your feedback. The rotational capacity of dipoles in the direction of the in-plane (IP) electric field is better, as the energy required to induce electric polarization rotation is lower compared to the out-of-plane (OOP) electric field [R1]. The more the dipole rotates, the greater the overall remnant polarization, thereby creating a wider memory window when subjected to the in-plane electric field.

The ferroelectric memory window is influenced not only by material parameters like remnant polarization (P_r) and coercive voltage (V_c), but also by structural parameters such as interlayer dielectric thickness, gate electrode position, interlayer thickness, and ferroelectric material thickness. Our investigation focused on the impact of ferroelectric material thickness and gate electrode position. We observed that as the ferroelectric material thickness increases and the distance between the channel and the gate electrode decreases, the memory window tends to widen. These values can be optimized by carefully designing the parameters. We have included this relevant information in Supplementary Figs. S7/S8 and the main article.

In the main article:

“...the difference in memory windows between the two cases became even more pronounced. **We observed that as the ferroelectric material thickness increases and the distance between the channel and the gate electrode decreases, the memory window tends to widen. Even at a distance of 30 μ m, the memory window obtained through the lateral gate remained superior to that achieved through the vertical gate. We have included the relevant information in Supplementary Figs. S7 and S8.**”

“Supplementary Figure S7 a) and b) are the images of LG-FeFET devices. Each set shares the channel (MoS₂) and dielectric (*h*-BN) materials to minimize the variations caused by the TMD flakes. c) and d) represent the I_d - V_g transfer curves of set-01 and set-02, respectively. e) illustrates the memory windows for various thicknesses.”

“Supplementary Figure S8 a) Optical microscopy image of the LG-FeFET device which has three different gate lengths. b) I_d - V_g transfer curves of the vertical and lateral gates and c) the memory windows. The lateral gates are positioned at distances of 10 μ m, 20 μ m, and 30 μ m away from the channel area, respectively. All curves show counterclockwise hysteresis.”

The electric field across the ferroelectric layer was calculated by dividing the voltage dropped in each layer by their thickness (for the vertical gate) or length (for the lateral gate). We have added the relevant information to the main article.

In the main article:

“...without structural effects. The electric field across the ferroelectric layer was calculated by dividing the voltage dropped in each layer by their thickness (for the vertical gate) or length (for the lateral gate).”

In this study, it was verified that the presence of an interlayer (SiO₂) in the LG-FeFET device leads to a broader memory window compared to the configuration without an interlayer, as depicted in Fig. S10. While the reviewer mentioned that the interlayer (SiO₂) would increase

gate voltage, it actually serves the purpose of preventing carrier injection from the metal gate, thereby preserving the remnant polarization. In the case of a directly contacted vertical gate, the injected screen charges into the ferroelectric layer compensate for the polarization charges by inducing a depolarization field. This is because α -In₂Se₃ is a semiconducting ferroelectric material with a bandgap of 1.3 eV.

The energy diagrams presented in Figs. 3g and 3h are conceptual representations based on both the Landau-Devonshire theory and first-principles DFT calculations [R1,R2]. We have revised the caption of Fig. 3.

In the caption of Fig.3:

“... The energy diagrams based on the Landau-Devonshire theory and first-principle density functional theory calculation, with respect to (g) the OOP and (h) the IP directional electric fields^{34,60}.”

Comment 3: How reproducible are the reported characteristics of the device? The authors are required to show the multi cycle sweep on multiple devices.

Author response 3: The reproducibility was confirmed using a total of eight distinct LG-FeFET devices. Across all devices, a consistent counterclockwise hysteresis was observed in the I_d - V_g transfer characteristic curves throughout 100 double-sweep cycles. However, slight variations in the characteristic curves were present due to fabrication process discrepancies and variations in the flakes. We have included additional details regarding this information in Supplementary Fig. S3 and the main article.

“Supplementary Figure S3 a) I_d - V_g transfer curves and b) memory windows for eight different LG-FeFET devices throughout 100 double-sweep cycles.”

In the main article:

“... the applied in-plane directional electric field successfully reverses the directions of polarization in α -In₂Se₃ layers. The reproducibility of the ferroelectric operation was confirmed using a total of eight distinct LG-FeFET devices. Across all devices, a consistent counterclockwise hysteresis was observed in the I_d - V_g transfer characteristic curves throughout 100 double-sweep cycles (Supplementary Fig. S3).”

Comment 4: For the retention measurement, it is unclear why there is no direct transistor state retention measurement, but rather the PFM. The authors need to show the transistor measurement results.

Author response 4: We appreciate your valuable feedback. To observe the behavior of out-of-plane (OOP) and in-plane (IP) polarizations, we utilized PFM measurements. This approach was necessary because the inter-coupling effect cannot be measured in an integrated transistor. The retention of an LG-FeFET device, as measured below, exhibits a shorter retention time compared to that of individual flakes predicted through PFM analysis. Several factors, including measuring conditions, neighboring layers, and defects, can account for this difference. To enhance retention, it is crucial to carefully optimize the thickness of the ferroelectric and dielectric layers, as well as the program/erase/read pulses. However, since our primary focus in this paper was to propose the concept of the LG-FeFET, we did not extensively explore optimization studies. We have included this information in the main article.

“Supplementary Figure S11 Retention characteristics of an LG-FeFET device. “

In the main article:

“... The OOP and IP polarization were simultaneously monitored over time to explore the intercoupling effect on the retention characteristic. The retention characteristic was evaluated using PFM, as direct measurement of the inter-coupling effect in an integrated transistor is difficult, where the inner square ...”

“... between polarizations. The LG-FeFET, as shown in Supplementary Fig. S11, is expected to exhibit a shorter retention time compared to that predicted on individual flakes through PFM analysis. Several factors, including measuring conditions, neighboring layers, and defects, can account for this difference. The LG-FeFET device’s endurance ...”

Comment 5: A big issue with this work is the lack of experimental details regarding the electrical measurements. Are all the measurements done in DC? What is the switching time of the device? How about the gate leakage current? What are the write pulses for the synaptic demonstration and endurance measurement?

Author response 5: We apologize for any confusion caused. All electrical measurements were conducted while applying pulse voltage. We performed separate optimization of the incremental step pulse program/erase (ISPP/ISPE) conditions for the vertical and lateral gates. For the ISPP condition, we set the start voltage at 2.3 V and the stop voltage at 4.0 V, with an increment of 13 mV. The ISPE condition, on the other hand, had a start voltage of -3.0 V and a

stop voltage of -4.0 V, with an increment of 8 mV. Both pulse rates were maintained at 1 kHz. The states were verified at a gate voltage of 0.7 V after each program/erase pulse.

In the main article:

“... both vertical and lateral gates. The incremental step pulse program/erase (ISPP/ISPE) conditions were separately optimized for the vertical and lateral gates. For the ISPP condition, we set the start voltage at 2.3 V and the stop voltage at 4.0 V, with an increment of 13 mV. The ISPE condition, on the other hand, had a start voltage of -3.0 V and a stop voltage of -4.0 V, with an increment of 8 mV. Both pulse rates were maintained at 1 kHz. The states were verified at a gate voltage of 0.7 V after each program/erase pulse.”

During the endurance test, the program and erase pulses were applied with amplitudes of 4 V and -7 V, respectively. After each pulse, the programmed/erased states were verified under a read voltage of -0.7 V. The pulse rate for the endurance test was set at 0.1 kHz. We have included this additional information in the main article.

In the main article:

“...ferroelectric material fatigue. During the endurance test, the program and erase pulses were applied with amplitudes of 4 V and -7 V, respectively. After each pulse, the programmed/erased states were verified under a read voltage of -0.7 V. The pulse rate for the endurance test was set at 0.1 kHz.”

Due to limitations in our measurement equipment, our investigation of high-frequency ferroelectric switching was constrained. However, the ferroelectric switching time of the α -In₂Se₃ ferroelectric device has been previously reported in reference [R3]. According to the reference, the polarization of α -In₂Se₃ is known to be switched in as low as 40 ns. We have included this information in the main article.

In the main article:

“...at room temperature in the ultrathin scale³⁴⁻³⁸. Previous studies have demonstrated the presence of ferroelectricity in a single layer (approximately 1.3 nm) of α -In₂Se₃ and the intercoupling effect in tri-layers (approximately 3 nm)^{39, 40}. An α -In₂Se₃ FeFET has shown a remarkably fast ferroelectric switching time as low as 40 ns⁴¹.”

Finally, as illustrated below, the gate leakage was negligible due to the insertion of the h-BN layer between the channel and the α -In₂Se₃ layers.

Comment 6: How to make sense of Fig.5d is unclear. Whether a linear result is obtained for MAC is unclear. The middle panel for Fig.5d should be W1F.

Author response 6: Appreciate your feedback. The triangular shapes indicate the input voltage signals at the ILs, which do not mean the weights for MAC. As shown in Fig. 5d, two LG-FeFET devices are linked to the CSL line, from which the output currents come out. Different input voltages were applied to the 1F and 2F ILs of LG-FeFET devices, each with distinct weight values, resulting in different CSL currents. The currents correspond to the product of the input signal amplitudes and the respective weight values. Subsequently, these currents are accumulated at the CSL, following Kirchhoff's law, as the sources of the two LG-FeFET devices are interconnected at a shared node. We have revised Fig. 5d and incorporated additional relevant information into the main article.

In the main article:

“...in two stacked LG-FeFET devices (Fig. 5c). As shown in Fig. 5d, two LG-FeFET devices are linked to the CSL line, from which the output currents come out. Different input voltages were applied to the 1F and 2F ILs of LG-FeFET devices, each with distinct weight values, resulting in different CSL currents. The currents correspond to the product of the input signal amplitudes and the respective weight values. Subsequently, these currents are accumulated at the CSL, following Kirchhoff's law, as the sources of the two LG-FeFET devices are interconnected at a shared node. Refer to the two cases in Fig. 5d.”

Comment 7: The authors need to discuss about the scalability of the lateral FeFET.

Author response 7: As detailed in the main article, the LG-FeFET presents several benefits, particularly in terms of reducing vertical height, since it does not require a metal gate in that direction. This reduction in overall stack height leads to a decrease in the total vertical resistance of the interconnection lines. With the height of each layer reduced, it becomes easier to stack more layers during fabrication. As a result, the occupied area can be minimized while maintaining the same cell density, as depicted in Fig. 5e.

References

- [R1] Ding, W. et al. Prediction of intrinsic two-dimensional ferroelectrics in In₂Se₃ and other III₂-VI₃ van der Waals materials. *Nat. Commun.* **8**, 14956 (2017).
- [R2] Devonshire AF. XCVI. Theory of barium titanate. *Philosophical Magazine Series 1* **40**, 1040-1063 (1949).

[R3] Wang S, *et al.* Two-dimensional ferroelectric channel transistors integrating ultra-fast memory and neural computing. *Nature Communications* **12**, 53 (2021).

Reviewer #3 (Remarks to the Author):

This paper investigates the laterally gated FeFET using α - In_2Se_3 as the ferroelectric, the interlocking of IP and OOP, the memory characteristics of the device and finally its application for in-memory computing. The work is original, interesting, scientifically sound and I recommend it for publication. Here are my comments:

Author response: Thank you for reviewing our paper. We appreciate your insightful comments and revised the manuscript accordingly.

Please find below our responses (in blue) to each of your specific comments (in black). Revisions to the original article are indicated in red.

Comment 1: Could you briefly elaborate on the size effect of the interlocking effect, i.e. what happens when the vertical and lateral size of the ferroelectric (FE) layer change? E.g. one would expect that the effect of the lateral gate is stronger for thinner and laterally shorter FE layers.

Author response: Thank you for your feedback. As the thickness increases and the distance between the channel and the gate electrode decreases, the memory window tends to widen. This phenomenon can be attributed to the varying strength of the applied electric field and the depolarization field. Specifically, the depolarization field weakens as the thickness increases, while the applied electric field strengthens as the length becomes shorter. We have included these pertinent findings in Supplementary Figs. S7 & S8, and the main article.

“Supplementary Figure S7 a) and b) are the images of LG-FeFET devices. Each set shares the channel (MoS_2) and dielectric (h -BN) materials to minimize the variations caused by the TMD flakes. c) and d) represent the I_d - V_g transfer curves of set-01 and set-02, respectively. e) illustrates the memory windows for various thicknesses.”

“Supplementary Figure S8 a) Optical microscopy image of the LG-FeFET device which has three different gate lengths. b) I_d - V_g transfer curves of the vertical and lateral gates and c) the memory windows. The lateral gates are positioned at distances of 10 μm , 20 μm , and 30 μm away from the channel area, respectively. All curves show counterclockwise hysteresis.”

In the main article:

“...the difference in memory windows between the two cases became even more pronounced. We observed that as the ferroelectric material thickness increases and the distance between the channel and the gate electrode decreases, the memory window tends to widen. Even at a distance of 30 μm , the memory window obtained through the lateral gate remained superior to that achieved through the vertical gate. We have included the relevant information in Supplementary Figs. S7 and S8.”

Comment 2: If this is the case, does this pose a limitation for the future device size and architectures? Because, it is well known that the thicker the FE layer, the larger the memory window (e.g. check [R1] on this). However, if there is a maximum thickness above which the lateral gate cannot switch the polarization and/or the interlocking is not efficient, this might be a problem.

Author response 2: We appreciate your insightful remarks. According to the reviewer's feedback, the α - In_2Se_3 in the LG-FeFET might have an optimum thickness similar to the case of hafnium-based ferroelectric materials [R1,R2]. Nonetheless, the ferroelectric switching behavior resulting from the inter-coupling effect of α - In_2Se_3 has been verified across a broad range, spanning from 2.3 nm (bilayer) to 130 nm (bulk), as evident in previous research [R1-R7] and our own investigation. This is because the spontaneous polarization of α - In_2Se_3 is triggered by its atomic configuration, not the crystal phase which causes polarization in hafnium-based ferroelectric materials.

In the main article:

“...at room temperature in the ultrathin scale³⁴⁻³⁸. Previous studies have demonstrated the presence of ferroelectricity in a single layer (approximately 1.3 nm) of α - In_2Se_3 and the intercoupling effect in tri-layers (approximately 3 nm)^{39,40}. An α - In_2Se_3 FeFET has shown a remarkably fast ferroelectric switching time as low as 40 ns⁴¹.”

Comment 3: On the other hand, the FE layer is relatively thick (60-70nm) as compared to the sizes of highly scaled memory devices in the semiconductor industry. Can this be further scaled down? What is the prospect?

Author response 3: Scaling down the thickness of the ferroelectric α -In₂Se₃ layer is indeed viable, as indicated by prior research, where it has been demonstrated that α -In₂Se₃ displays ferroelectric properties even in a single layer (approximately 1.3 nm) and the intercoupling effect becomes evident in tri-layers (approximately 3 nm) [R3,R7].

Of course, when designing a FeFET device with a thin ferroelectric film, two crucial factors should be taken into account: i) the thickness of the ferroelectric layer and ii) the interface between the ferroelectric layer and the electrode/channel region. This is because such factors can lead to the potential challenges associated with depolarization field. Nonetheless, recent advancements and increased focus on thin-film growth technology for vdW ferroelectric materials would facilitate the application of vdW ferroelectric films toward industrial semiconductor devices.

Comment 4: What are the main ferroelectric properties of the FE layer (e.g. coercive field, polarization etc)? Could you please provide this in the main text?

Author response 4: Obtaining such properties of α -In₂Se₃ using an MFM capacitor is challenging due to its distinctive semiconducting property (with an energy bandgap of approximately 1.3 eV). Previous research has reported the coercive voltages (V_c) of α -In₂Se₃ within the range of 1.7 V to 5.5 V through PFM analysis [R4, R5, R8-R11], and our own investigations also revealed a similar value of 1.76 V, as illustrated in supplementary Fig. S3.

As of now, there is no established methodology to extract the remnant polarization charges (P_r) from the LG-FeFET device for α -In₂Se₃ with semiconducting properties. To accurately determine P_r , it needs to be further divided into in-plane and out-of-plane components. Additionally, it is essential to separate the contribution of carriers in α -In₂Se₃ from the P_r values for a proper understanding of the polarization behavior. Thus, further comprehensive and in-depth studies are required.

We have added the relevant information to the main article.

In the main article:

“... The local phase loop and the amplitude are in supplementary Fig. S3. **Obtaining ferroelectric properties of α -In₂Se₃ using an MFM capacitor is challenging due to its distinctive semiconducting properties (with an energy bandgap of approximately 1.3 eV). Here, we revealed a coercive voltage (V_c) of 1.76 V through the PFM analysis. Secondly, ...”**

Comment 5: Fig. 4(i) shows the cycling endurance. What amplitude/pulse duration was applied? Please provide in the main text. In general, how does the device behave under pulsed operation? What is the switching time?

Author response 5: During the endurance test, the amplitudes of the program and erase pulses are 4 V and -7 V, respectively, with a frequency of 0.1 kHz. Following each pulse, the programmed/erased states are confirmed using a read voltage of -0.7 V.

Generally, 0 V is applied to the source/drain, while a program/erase pulse is applied to the gate. Following a program/erase operation, a read operation is typically performed to confirm the states. Our investigation of high-frequency ferroelectric switching was constrained by the capabilities of our measurement equipment. Nonetheless, previous research in [R12] reported the fast-switching of the α -In₂Se₃ FeFET, where its polarization could be switched within as little as 40 ns. We have included this valuable information in the main article.

In the main article:

“...ferroelectric material fatigue. During the endurance test, the program and erase pulses were applied with amplitudes of 4 V and -7 V, respectively. After each pulse, the programmed/erased states were verified under a read voltage of -0.7 V. The pulse rate for the endurance test was set at 0.1 kHz.”

“...at room temperature in the ultrathin scale³⁴⁻³⁸. Previous studies have demonstrated the presence of ferroelectricity in a single layer (approximately 1.3 nm) of α -In₂Se₃ and the intercoupling effect in tri-layers (approximately 3 nm)^{39,40}. An α -In₂Se₃ FeFET has shown a remarkably fast ferroelectric switching time as low as 40 ns⁴¹.”

Comment 6: The memory window, larger than 9V, as well as the retention and the endurance of this device are remarkable. However, it would be quite useful for the ferroelectrics/memory community to have a comparison with state-of-the art FeFETs, which are currently mainly made of Hafnium-(Zirconium)-Oxide. I suggest to provide a brief comparison in the main text, e.g., in terms of the main device and material properties summarized in [R1].

Author response 6: Thank you for your comments. Before comparing the properties of the LG-FeFET using α -In₂Se₃ with hafnium-oxide-based FeFET devices, we note that these properties can be varied by physical dimension and measurement conditions.

We added a brief comparison table between the HZO-based FeFET and LG-FeFET in Supplementary Table 2 as below. Most of all, the LG-FeFET is distinguished from HZO-based FeFET in the direction of the applied electric field. The LG-FeFET exhibits a larger memory window (approximately 10 V) compared to HZO FeFETs (smaller than 5 V). The endurance of LG-FeFET was confirmed over 10^5 cycles which is comparable with HZO FeFETs, but the retention is shorter than that of HZO-based FeFETs. Due to the unique semiconducting properties of α -In₂Se₃ ($E_g \approx 1.3$ eV), the coercive field (E_c) and the remnant polarization (P_r) cannot be directly compared.

In the Supplementary:

Device	Hafnium-oxide based FeFET	LG-FeFET
Typical materials	X:HfO ₂ , HZO	α -In ₂ Se ₃
Deposition	ALD, PVD	Exfoliation
Direction	Out-of-plane (OOP)	In-plane (IP)

Thickness	5 – 20 nm	40 – 130 nm
Memory window	1.5 – 3.5 V	About 10 V
Endurance	$<10^8$ cycles	$> 10^5$ cycles
Retention	$>10^6$ s	About 10^4 s (Flake) About 10^2 s (Device)
References	[R1]	This work

“Supplementary Table 2. Comparison of the properties between an HZO-based FeFET and α -In₂Se₃ based LG-FeFET”

In the main article:

“... the programmed (low conductance) and erased (high conductance) states. We briefly compared the features of LG-FeFET and HZO-based FeFET in Supplementary Table 2. The LG-FeFET exhibits a larger memory window (approximately 10 V) compared to HZO-based FeFETs (< 5 V)⁶⁴. The endurance of LG-FeFET is comparable with HZO FeFETs, but the retention is shorter than that of HZO FeFETs. Due to the unique semiconducting properties of α -In₂Se₃ ($E_g \approx 1.3$ eV), the coercive field (E_c) and the remnant polarization (P_r) cannot be directly compared.”

References

- [R1] Mulaosmanovic, H., et al. Ferroelectric field-effect transistors based on HfO₂: a review. *Nanotechnology* 32.50 (2021): 502002.
- [R2]. Park SH, Kim JY, Song JY, Jang HW. Overcoming Size Effects in Ferroelectric Thin Films. *Advanced Physics Research*, 2200096 (2023).
- [R3] Xiao, J. et al. Intrinsic Two-Dimensional Ferroelectricity with Dipole Locking. *Phys. Rev. Lett.* **120**, 227601 (2018).
- [R4] Cui, C. et al. Intercorrelated In-Plane and Out-of-Plane Ferroelectricity in Ultrathin Two-Dimensional Layered Semiconductor In₂Se₃. *Nano Lett.* **18**, 1253-1258 (2018).
- [R5] Li, Y. et al. Orthogonal Electric Control of the Out-Of-Plane Field-Effect in 2D Ferroelectric α -In₂Se₃. *Adv. Electron. Mater.* **6**, 2000061 (2020).
- [R6] Dutta, D., Mukherjee, S., Uzhansky, M. & Koren, E. Cross-field optoelectronic modulation via inter-coupled ferroelectricity in 2D In₂Se₃. *npj 2D Mater. Appl.* **5**, 44934 (2021).
- [R7] Xue, F. et al. Room-Temperature Ferroelectricity in Hexagonally Layered α -In₂Se₃ Nanoflakes down to the Monolayer Limit. *Adv. Funct. Mater.* **28**, 1803738 (2018).
- [R8] Si, M. et al. Asymmetric Metal/ α -In₂Se₃/Si Crossbar Ferroelectric Semiconductor Junction. *ACS Nano* **15**, 5689-5695 (2021).
- [R9] Baek S, et al. Ferroelectric Field-Effect-Transistor Integrated with Ferroelectrics Heterostructure. *Advanced Science* **9**, 2200566 (2022).
- [R10] Park, S., Oh, S., Lee, D. & Park, J.-H. Ferro-floating memory: Dual-mode ferroelectric floating memory and its application to in-memory computing. *InfoMat* **4**, e12367 (2022).
- [R11] Si, M. et al. A ferroelectric semiconductor field-effect transistor. *Nat. Electron.* **2**, 580-586 (2019).
- [R12] Wang S, et al. Two-dimensional ferroelectric channel transistors integrating ultra-fast memory and neural computing. *Nature Communications* **12**, 53 (2021).

REVIEWERS' COMMENTS

Reviewer #1 (Remarks to the Author):

The authors have clarified the significance of this paper more explicitly in the revised version and have conducted a series of compelling experiments. Therefore, I am pleased to recommend accepting this paper as it is.

Reviewer #3 (Remarks to the Author):

The authors have addressed all my questions and comments.

The work is novel and original and therefore I recommend it for publication.